# Single molecule magnet with an unpaired electron trapped between two lanthanide ions inside a fullerene

Fupin Liu[1,*], Denis S. Krylov[1,*], Lukas Spree[1], Stanislav M. Avdoshenko[1], Nataliya A. Samoylova[1], Marco Rosenkranz[1], Aram Kostanyan[2], Thomas Greber[2], Anja U.B. Wolter[1], Bernd Büchner[1] & Alexey A. Popov[1]

Increasing the temperature at which molecules behave as single-molecule magnets is a serious challenge in molecular magnetism. One of the ways to address this problem is to create the molecules with strongly coupled lanthanide ions. In this work, endohedral metallofullerenes $Y_2@C_{80}$ and $Dy_2@C_{80}$ are obtained in the form of air-stable benzyl monoadducts. Both feature an unpaired electron trapped between metal ions, thus forming a single-electron metal-metal bond. Giant exchange interactions between lanthanide ions and the unpaired electron result in single-molecule magnetism of $Dy_2@C_{80}(CH_2Ph)$ with a record-high 100 s blocking temperature of 18 K. All magnetic moments in $Dy_2@C_{80}(CH_2Ph)$ are parallel and couple ferromagnetically to form a single spin unit of 21 $\mu_B$ with a dysprosium-electron exchange constant of 32 cm$^{-1}$. The barrier of the magnetization reversal of 613 K is assigned to the state in which the spin of one Dy centre is flipped.

[1] Leibniz Institute for Solid State and Materials Research, Helmholtzstrasse 20, 01069 Dresden, Germany. [2] Physik-Institut der Universität Zürich, Winterthurerstrasse 190, CH-8057 Zürich, Switzerland. * These authors contributed equally to this work. Correspondence and requests for materials should be addressed to A.A.P. (email: a.popov@ifw-dresden.de).

The discovery of single-molecule magnetism (SMM) in the Mn$_{12}$ complex in 1993 (ref. 1) opened the perspectives for SMM in information storage, molecular spintronics and quantum computing[2]. Lanthanides entered the field in 2003 with the report on the slow relaxation of magnetization in their double-decker complexes[3] and hundreds of lanthanide SMMs were described since that time[4–10]. The increase of the temperature, at which molecules behave as SMMs, still remains one of the main challenges. Several strategies were developed to reduce the rate of the quantum tunnelling of magnetization (QTM), which is one of the main relaxation mechanisms in SMMs in zero field. One approach implies the creation of a highly symmetric environment around lanthanide ions[11–15]. Another strategy is based on the coupling of two or more lanthanide ions with a large spin ground state[4,16–18]. In polynuclear systems, the exchange barrier reduces zero-field QTM and completely changes the relaxation dynamics. Endohedral metallofullerenes (EMFs) DySc$_2$N@C$_{80}$ and Dy$_2$ScN@C$_{80}$ provide an illustrative example: The former exhibits fast QTM in zero field[19], whereas the latter shows pronounced remanence[20]. SMMs with radical bridges between lanthanides were shown to give the highest blocking temperature among coupled systems[21]. A very strong superexchange is achieved in [M–N$_2^{3-}$–M] complexes due to antiferromagnetic coupling of lanthanide ions with the N$_2^{3-}$ radical bridge (M = Dy, Tb)[16,22]. Yet, the potential of the strong exchange coupling for rare-earth-based SMMs is far from being fully explored.

The ultimate realization of exchange coupling might be achieved when two lanthanides share a single-electron covalent bond. However, in a recent monograph on metal–metal bonding, not a single example of a lanthanide–lanthanide bond is mentioned[23]. Lanthanides tend to give their valence electrons away and make compounds with largely ionic bonding. Yet, encapsulation of lanthanide atoms inside a carbon cage creates a suitable environment for the formation of lanthanide–lanthanide bonds[24]. Although metal atoms in EMFs transfer their valence electrons to carbon cages[25], the M–M bonding molecular orbital (MO) with spd-hybrid character is one of the frontier MOs in dimetallofullerenes[24].

Detailed analysis of the M–M bonding in dimetallofullerenes revealed that this phenomenon is related to the energy of the $(ns)\sigma_g^2$ MO of the respective M$_2$ dimers[24,26]. In Lu$_2$, the Lu–Lu $(6s)\sigma_g^2$ orbital has relatively low energy, which hence remains occupied when Lu$_2$ is placed inside a fullerene cage. On the other side of the lanthanide row, La has a high energy of the $(6s)\sigma_g^2$ MO in the La$_2$ dimer and in La-dimetallofullerenes the electrons are fully transferred from this MO to a carbon cage, leading to the La$^{3+}$ state without La–La bonding[27]. Y and medium-size lanthanides (such as Gd or Dy) with intermediate values of the metal–metal $(ns)\sigma_g^2$ MO may have different bonding situations depending on the energy match between MOs of the hosting fullerene cage and the metal–metal bonding MO.

The C$_{80}$-$I_h$ fullerene with threefold degenerate lowest unoccupied molecular orbital (LUMO) naturally acts as a six-electron acceptor in different types of EMFs. For instance, in the La$_2$@C$_{80}$-$I_h$, each metal ion is charged 3 + and the La–La orbital in La$_2$@C$_{80}$ is the LUMO[27]. However, for Y or lanthanides with higher energy of the M–M bonding MO than in La$_2$, computations by Shinohara and colleagues[28] showed that the M$_2$@C$_{80}$-$I_h$ molecule has only one unpaired electron occupying the M–M bonding MO, the other electron being delocalized over the carbon cage. The formal oxidation state of metal atoms in M$_2$@C$_{80}$-$I_h$ molecule is thus + 2.5. Such molecules are unstable radicals and still remain elusive, although their existence has been demonstrated by transforming into more chemically stable forms (anions[29,30] or derivatives[28,31,32]). A single-electron M–M bond has been also stabilized by a substitution of one carbon atom by nitrogen, giving azafullerenes M$_2$@C$_{79}$N (M = Y, Gd, Tb)[33,34]. Computational studies predicted a large ferromagnetic coupling of Gd ions in Gd$_2$@C$_{79}$N (refs 35,36) and unusual magnetic properties in Dy$_2$@C$_{79}$N (ref. 35).

In this work we describe elusive M$_2$@C$_{80}$-$I_h$ (M = Y, Dy) species obtained as air-stable chemical derivatives, benzyl monoadducts M$_2$@C$_{80}$(CH$_2$Ph). The hyperfine structure in

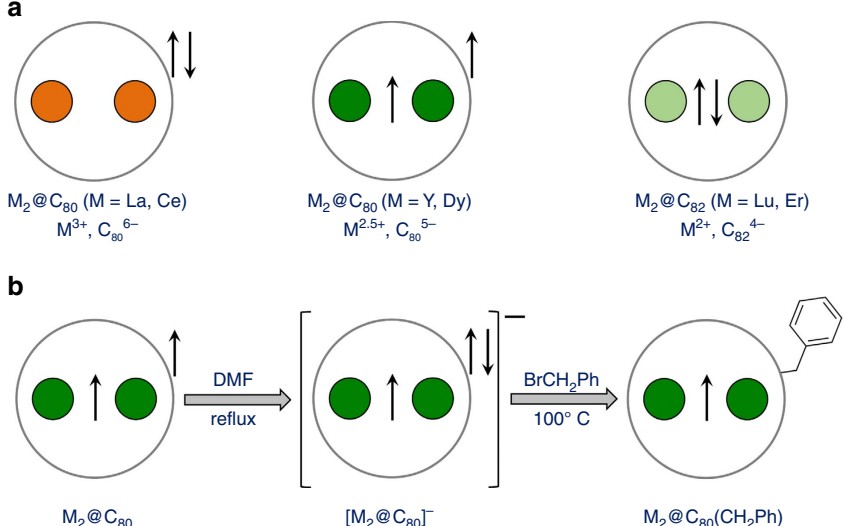

**Figure 1 | Schematic depiction of dimetallofullerenes and synthetic route to M$_2$@C$_{80}$(CH$_2$Ph) derivatives. (a)** Comparison between different types of dimetallofullerenes: in La$_2$@C$_{80}$ each metal atoms is trivalent, transfers three electrons to the cage and the M–M bonding MO is the LUMO; in M$_2$@C$_{80}$ (Y, Dy) studied in this work, each metal formally transfers 2.5 electrons to the cage and the M–M bonding MO is single occupied; in M$_2$@C$_{82}$ (Er,Lu), metals transfer two electrons each and form two-electron M–M covalent bond. **(b)** Description of the synthetic route for M$_2$@C$_{80}$ derivative with single-electron M–M bond developed in this work (M = Y, Dy): M$_2$@C$_{80}$ molecules synthesized by arc-discharge are present in triplet state in the soot; upon reduction with DMF, soluble mono-anions are formed, the surplus electron goes to the cage and the M–M SOMO is preserved. At the next stage, anions are reacted with benzyl bromide, which leads to stable non-charged mono-adducts. These adducts are further separated by HPLC.

the electron paramagnetic resonance (EPR) spectra of $Y_2@C_{80}(CH_2Ph)$ reveals the presence of the single-electron metal–metal bond with strong interaction of the unpaired electron spin with metal atoms. In $Dy_2@C_{80}(CH_2Ph)$, the single-electron Dy–Dy bond leads to SMM with record-high blocking temperature of magnetization.

## Results

**Synthesis and molecular structure.** Dy- and Y-EMFs are synthesized using standard metal oxide/graphite arc-discharge synthesis and processed as depicted in Fig. 1. Extraction of fullerenes from the soot with dimethylformamide (DMF) gives a mixture of anionic monometallofullerenes and dimetallofullerenes (Supplementary Fig. 1). The solubility of empty fullerenes in DMF is very low, whereas the selectivity of this solvent towards EMFs is due to the formation of well-soluble anions[37–39]. EPR spectroscopy proves the presence of a single-occupied Y–Y bonding MO in the $Y_2@C_{2n}^-$ anions in DMF solution. The localization of spin density on two Y atoms results in a 1:2:1 triplet with a large $a(^{89}Y)$ hyperfine coupling constant (81.2 G in $Y_2@C_{79}N$ (ref. 33)). The DMF extract of Y-EMFs exhibits three such triplets, with $a(^{89}Y)$ constants of 64.5, 72.1 and 76.2 G (Supplementary Fig. 2 and Supplementary Note 1). Presumably, they correspond to $Y_2@C_{78}^-$ and two isomers of $Y_2@C_{80}^-$ with $I_h$ and $D_{5h}$ cage symmetry.

The air-stable neutral molecular form of the compounds was obtained by reaction of $M@C_{2n}^-/M_2@C_{2n}^-$ (M = Y, Dy) monoanions in DMF with benzyl bromide, yielding a series of benzyl monoadducts. EPR proves that characteristic features of the Y–Y bonding single-occupied molecular orbital (SOMO) are preserved in the $Y_2@C_{78,80}(CH_2Ph)$ species after the derivatization (Supplementary Fig. 3 and Supplementary Note 1). Despite being radicals, the $M_2@C_{78,80}(CH_2Ph)$ derivatives are air stable, do not show signs of degradation over several months and thus do not require special handling conditions. The mixtures of EMF derivatives was then subjected to multi-step high-performance liquid chromatography (HPLC) separation (Supplementary Figs 4–10 and Supplementary Note 2), which led to the isolation of pure $Y_2@C_{80}(CH_2Ph)$ and $Dy_2@C_{80}(CH_2Ph)$ denoted hereafter as **$Y_2$-I** and **$Dy_2$-I**.

Figure 2a,b shows the molecular structure of **$Dy_2$-I** determined by single crystal X-ray diffraction (further details are given in Supplementary Figs 11–13, Supplementary Table 1 and Supplementary Note 3). The benzyl group is attached to the carbon on the pentagon/hexagon/hexagon junction of the $I_h(7)$-$C_{80}$ fullerene cage (Fig. 2a). Dy atom occupies two positions with fractional occupancies near 0.7 and 0.3 (Fig. 2b). The $C_{80}(CH_2Ph)$ moiety has $C_s$ symmetry and the two pairs of Dy positions are related via the mirror plane operation. Dy–Dy distances in the two pairs are 3.896(1) and 3.898(3) Å. Dy atoms are coordinated to hexagons of the fullerene cage in a quasi-$\eta^6$ manner with Dy–C distances in the coordinated hexagon between 2.308(8) and 2.586(9) Å.

$C_{80}$-$I_h$ cage has only two types of carbons, leading to two possible $C_{80}(CH_2Ph)$ isomers, and multiple isomers of $M_2@C_{80}(CH_2Ph)$ due to different orientations of metal atoms inside the cage. Density functional theory (DFT) computational studies performed for $Y_2@C_{80}(CH_2Ph)$ isomers show that benzyl addition to the carbon atom on a pentagon/hexagon/hexagon junction is energetically more favourable by more than 60 kJ mol$^{-1}$ than the addition to a carbon on a triple hexagon junction. There are several almost isoenergetic positions for metal atoms within the cage (Supplementary Fig. 14), all distributed near the belt of hexagons highlighted yellow in Fig. 2a,b. The lowest energy conformer corresponds to the positions of metal

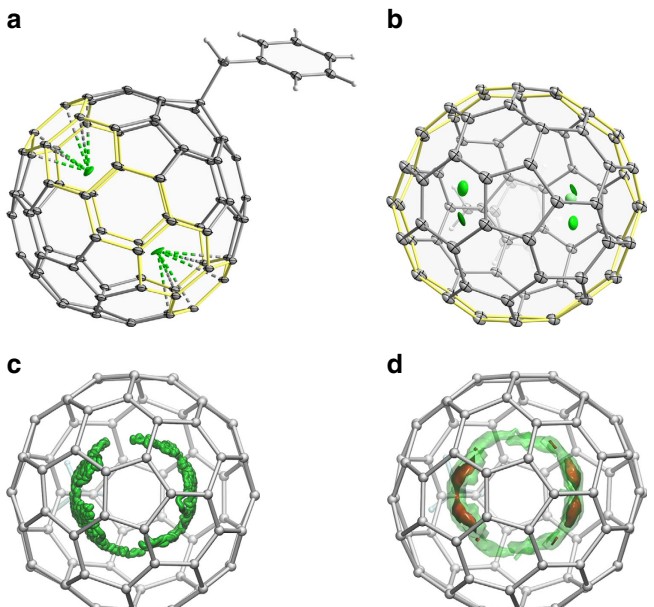

**Figure 2 | Molecular structure and internal dynamics in $M_2@C_{80}(CH_2Ph)$.** (**a**) Molecular structure of **$Dy_2$-I** from single-crystal X-ray diffraction shown with 50% thermal ellipsoids. Only the major positions of Dy atoms (occupancy near 70%) are shown, the belt of hexagons around which metal dimer can rotate is highlighted yellow. (**b**) Same as **a** but showing the molecule in a different orientation and with all Dy positions (green). (**c**) Molecular dynamics trajectory of $Y_2@C_{80}(CH_3)$ propagated for 60 ps at 300 K, Y atoms are shown green, displacements of carbon atoms are not shown for clarity. (**d**) Probability density to find metal atoms derived from molecular dynamics simulations and plotted with two isosurfaces (solid, isovalue 0.25, high probability; transparent, isovalue 0.025, lower probability; $C_s$ symmetry of the $C_{80}(CH_2Ph)$ moiety was taken into account in calculations of probability distribution). In **b–d**, the molecules are oriented so that the addend is located on the opposite side of the molecule.

atoms in the X-ray structure of **$Dy_2$-I**. The presence of several energy minima for metal atoms within the energy range of kJ mol$^{-1}$ suggests that the $M_2$ unit may rotate inside the fullerene.

DFT-based Born–Oppenheimer molecular dynamics (BOMDs) simulations of $Y_2@C_{80}(CH_3)$ were performed to analyse the dynamics of metal atoms (CH$_3$ group was used instead of benzyl group as the main focus was on the motion of metal atoms). The BOMD trajectory propagated for 60 ps at 300 K and probability distribution of metal atoms obtained from this trajectory are shown in Fig. 2c,d. Metal atoms are found to rotate within one plane with the largest probability density at the same positions as for Dy atoms in the X-ray structure of **$Dy_2$-I**.

**Spectroscopic and electrochemical properties.** The EPR spectrum of **$Y_2$-I** solution at room temperature (Fig. 3a) shows a triplet with the isotropic $a(^{89}Y)$ hyperfine constant 223.8 MHz (81.0 G) and a $g$-factor of 1.9733. The spectrum of the frozen solution at 150 K exhibits characteristic rhombic pattern with hyperfine tensor components of 208 and 246 MHz for $a_\perp(^{89}Y)$ and $a_\parallel(^{89}Y)$, respectively, and $g$-tensor components $g_\perp = 1.9620$ and $g_\parallel = 1.9982$. The large $a(^{89}Y)$ values and significant anisotropy of both tensors indicate that a substantial degree of the spin density is localized evenly on the two Y atoms. The EPR parameters of **$Y_2$-I** are very close to those reported earlier for $Y_2@C_{79}N$ (refs 33,40), proving that both molecules have very similar spin density distribution stemming from the single-occupied Y–Y bonding

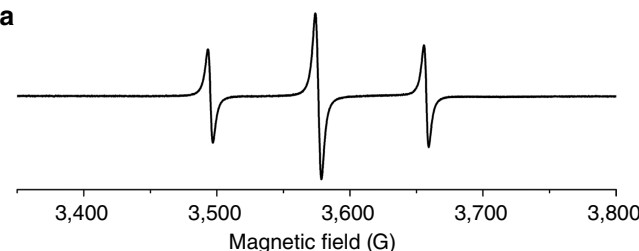

**a**

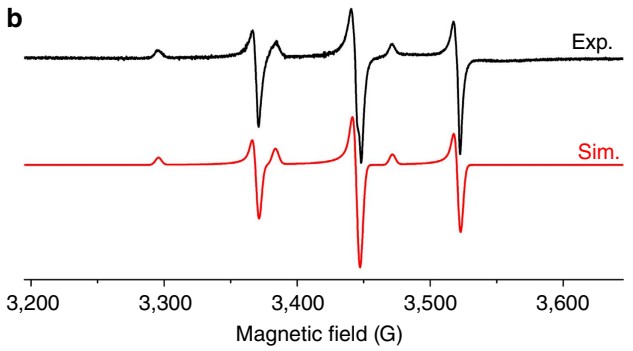

**b**

Exp.

Sim.

**Figure 3 | EPR spectroscopy of Y₂-I.** (**a**) EPR spectrum of **Y₂-I** in toluene solution at room temperature: $a(^{89}Y) = 223.8$ MHz, $g = 1.9733$. (**b**) Experimental and simulated EPR spectra of **Y₂-I** in frozen toluene solution at 150 K. Simulation parameters: $a_\perp(^{89}Y) = 208.0$ MHz, $a_\parallel(^{89}Y) = 245.9$ MHz, $g_\perp = 1.9620$ and $g_\parallel = 1.9982$.

MO. DFT calculations confirm that the spin density in **Y₂-I** is fully enclosed inside the carbon cage and resembles the spatial distribution of the Y–Y bonding MO (Supplementary Fig. 15). Ultraviolet–visible–near-infrared absorption, infrared and Raman spectra (Supplementary Figs 16–18 and Supplementary Note 4) of **Y₂-I** and **Dy₂-I** are very similar, confirming that the two compounds have an identical molecular structure.

The redox behaviour of **Y₂-I** and **Dy₂-I** is quite remarkable. Their first oxidation, and the second and third reduction potentials are virtually identical (Table 1), whereas the first reduction potentials differ by 0.08 V, **Y₂-I** being easier to reduce (Fig. 4a and Supplementary Fig. 19). The metal dependence of the first reduction potential indicates that the reduction is associated with the filling of the single-occupied M–M bonding orbital. On the other hand, metal-independent oxidation potentials evidence for the fullerene-based oxidation. In full agreement with experimental observations, DFT calculations show that the occupied component of the M–M bonding SOMO in **Y₂-I** is buried below the cage-based HOMO and hence not accessible for oxidation (Fig. 4b). On the other hand, the LUMO of **Y₂-I** is the unoccupied counterpart of the SOMO and this orbital accepts an electron in the first reduction step. Interestingly, the first oxidation and reduction potentials of **La₂-I** are substantially more negative than those of **Y₂-I** and **Dy₂-I**, whereas the second and the third reductions occur at similar potentials (Table 1)[41]. Our MO analysis (Fig. 4b and Supplementary Fig. 20) shows that the oxidation of **La₂-I** proceeds via the La–La SOMO, whereas reduction represents the fullerene-cage-based process. The switching of the redox mechanisms in **M₂-I** on going from Dy or Y to La is caused by the higher energy of the $(6s)\sigma_g^2$ MO in the La₂ dimer, which is inherited in dimetallofullerenes in the form of a high-energy La–La bonding orbital.

**Magnetic properties of Dy₂-I.** The magnetic properties of **Dy₂-I** were first assessed by SQUID magnetometry. Magnetization

**Table 1 | Redox potentials of M₂-I (M = Y, Dy, La).**

| EMF | Ox-II | Ox-I | Red-I | Red-II | Red-III | gap$_{EC}$ |
|---|---|---|---|---|---|---|
| **Y₂-I** | 0.98 | 0.52 | −0.52 | −1.29 | −1.60 | 1.04 |
| **Dy₂-I** | 0.98 | 0.52 | −0.60 | −1.28 | −1.58 | 1.12 |
| **La₂-I** * | | 0.15 | −0.92 | −1.34 | −1.64 | 0.97 |

EMF, endohedral metallofullerene.
Potentials are measured versus Fe(Cp)₂$^{+/0}$ pair, and are given in Volts.
*La₂-I values are from ref. 41.

curves of the powder sample, shown in Fig. 5a and Supplementary Fig. 21, exhibit hysteresis at temperatures from 1.8 to 21 K. The average magnetic moment at saturation is 10.5 $\mu_B$ per molecule (Supplementary Note 5). Taking into account the random orientation of crystallites in the powder, the magnetic moment along the main magnetization axis amounts to $20.9 \pm 0.3 \mu_B$ per molecule. The SMM behaviour is characterized by a blocking temperature of the magnetization ($T_B$) defined as the temperature of the maximum of the susceptibility of a zero-field cooled sample. As a non-equilibrium parameter, $T_B$ depends on the temperature sweep rate and is found to vary between 18.3 at 1 K min⁻¹, 21.9 K at 5 K min⁻¹ (Fig. 5b) to 22.9 K at 20 K min⁻¹ (Supplementary Fig. 22). These are the highest blocking temperatures for any lanthanide-based SMM so far. The previous highest $T_B$ value of 14 K was reported for the [Tb–N₂³⁻–Tb] complex with a N₂³⁻ radical bridge[42] and for the pentagonal bipyramidal Dy single-ion magnet[11] (temperature sweep rates were not reported).

Magnetization relaxation times $\tau_M$ of **Dy₂-I** are determined using a stretched exponential fitting of the relaxation curves below 22 K (Supplementary Fig. 23, Supplementary Tables 2 and 3, and Supplementary Note 6) and from the ac-susceptibility ($\chi''$) measurements between 23 and 33 K (Fig. 5c, Supplementary Fig. 24 and Supplementary Tables 4 and 5). Below 5 K, zero-field $\tau_M$ values reach the temperature-independent regime, which in single-ion magnets is usually associated with QTM. For the multicentre system such as **Dy₂-I**, zero-field QTM is less likely, as it requires simultaneous flip of the whole spin system. However, if the constituting spins are strongly coupled, they may behave as a single entity, so that QTM cannot be excluded, and we therefore cautiously denote the process as QTM-like relaxation. Dilution of **Dy₂-I** in polystyrene leads to much longer relaxation times (Supplementary Figs 25 and 26), which increase with cooling without the sign of levelling off and reach 2 months at 2 K (Fig. 5c). The influence of dilution on $\tau_M$ shows that intermolecular interactions are the main reason of the QTM-like behaviour. Alternatively, QTM-like relaxation can be switched off by a constant field of 0.4 T, leading to the relaxation time of about 1.5 years at 2 K (Fig. 5c). A temperature variation of the relaxation time enables the determination of the more universal SMM character-istic, the temperature at which the relaxation time is 100 s, $T_{B(100)}$[43]. For **Dy₂-I**, we obtain $T_{B(100)} = 18$ K (Fig. 5c), which is the highest temperature ever reported for a single-molecule magnet.

The whole set of $\tau_M$ values for the non-diluted sample is described by a combination of several relaxation processes:

$$\text{Zero-field: } \tau_M^{-1} = \tau_{QTM}^{-1} + CT^n + \tau_{01}^{-1} \exp(-U_1^{eff}/T) \\ + \tau_{02}^{-1} \exp(-U_2^{eff}/T) \tag{1}$$

$$\text{In-field: } \tau_M^{-1} = CT^n + \tau_{01}^{-1} \exp(-U_1^{eff}/T) + \tau_{02}^{-1} \exp(-U_2^{eff}/T) \tag{2}$$

Both equations were fitted simultaneously for zero-field and in-field relaxation times. The first term in equation (1) corresponds to the QTM-like process ($\tau_{QTM} = 3257$ s), the second

term describes the Raman relaxation ($C = 8.2 \times 10^{-10}\,\mathrm{s}^{-1}\,\mathrm{K}^{-n}$, $n = 4.9$), whereas the two last terms describe relaxation via two Orbach processes ($U_1^{\mathrm{eff}} = 40\,\mathrm{K}$, $\tau_{01} = 13.6\,\mathrm{s}$; $U_2^{\mathrm{eff}} = 613\,\mathrm{K}$, $\tau_{02} = 3.6 \times 10^{-12}\,\mathrm{s}$). In zero-field, temperature-independent QTM-like relaxation is the main process from 1.8 K up to 5 K, when it starts to compete with the low-barrier Orbach process. The latter is the dominant relaxation mechanism from 10 to 18 K, whereas above 20 K the Orbach process with the larger barrier takes over. In the field of 0.4 T, QTM-like relaxation is switched off (hence the QTM term is absent in equation (2)). This opens the possibility for the low-barrier Orbach relaxation process to extend its temperature range on the lower side to ca 3 K. At $T < 3\,\mathrm{K}$, the Raman mechanism starts to dominate (Supplementary Fig. 27). The low-temperature Orbach process has rather unusual parameters, and we also tried to fit our experimental data using only one Orbach process in equations (1) and (2), but could not obtain a satisfactory agreement (Supplementary Fig. 28 and Supplementary Note 7).

The [Dy$^{3+}$–$e$–Dy$^{3+}$] system in **Dy₂-I** can be described by the zero-field effective spin Hamiltonian:

$$\hat{H}_{\mathrm{tot}} = \hat{H}_{\mathrm{CF(Dy1)}} + \hat{H}_{\mathrm{CF(Dy2)}} - 2j_{\mathrm{Dy1,Dy2}}\hat{J}_{\mathrm{Dy1}} \cdot \hat{J}_{\mathrm{Dy2}}$$
$$- 2j_{\mathrm{Dy1},e}\hat{J}_{\mathrm{Dy1}} \cdot \hat{S}_e - 2j_{\mathrm{Dy2},e}\hat{J}_{\mathrm{Dy2}} \cdot \hat{S}_e \qquad (3)$$

where the first two terms describe the crystal-field single-ion anisotropy of the Dy ions and the last three terms describe the exchange and dipolar interactions between the two Dy centres, and between Dy centres and the unpaired electron spin. We will first describe *ab initio* computations for the single-ion crystal field (CF) parameters and then proceed to the discussion of the exchange interactions and the spectrum of the Hamiltonian.

The crystal-field parameters for each Dy centre were computed for the [**DyY-I**]$^-$ molecule at the CASSCF/SO-RASSI level of theory with the use of the SINGLE_ANISO module[44] as implemented in MOLCAS 8.0 (ref. 45) (*ab initio* calculations of the **Dy₂-I** molecule with two Dy centres and an unpaired spin are not feasible). Calculations showed that both Dy centres have easy-axis magnetic anisotropy (the magnetic ground state of Dy$^{3+}$ has a $J_z$ projection of $\pm 15/2$). The quantization axes are parallel and aligned along the metal–metal bond. The overall CF splitting in both centres amounts to $\sim 900\,\mathrm{cm}^{-1}$, whereas the first and second excited states are found near 230–280 and 390–409 cm$^{-1}$ (Supplementary Fig. 29, Supplementary Tables 6 and 7, and Supplementary Note 8).

The high-spin ground state of Dy$^{3+}$ in **Dy₂-I** is rather counterintuitive. The crystal field in other Dy-EMFs exhibiting SMM behaviour is dominated by negatively charged nitride[20] or carbide[46] ions at the short distance of Dy, which leads to the quasi-uniaxial crystal field with an easy-axis single-ion anisotropy and large CF splitting[47,48]. In dimetallofullerenes, positive charges of metal ions are not counterbalanced by non-metals and easy-plane anisotropy ($J_z = \pm 1/2$) might be expected. Indeed, if a $+3$ point charge is placed at the position of one of the Dy ions, both the point charge model and *ab initio* CASSCF calculations predict an easy-plane ground state for the remaining Dy$^{3+}$ ion. However, the Dy–Dy bond localizes additional electron density between two lanthanide ions. When an additional negative point charge is placed at the midpoint between the Dy$^{3+}$ ion and the positive charge, the situation changes severely. The crystal field for this model system scales with the charge and distance as $q/R^3$. Figure 5d visualizes the crystal field potential (defined as $q_p/R_p^3 + q_m/R_m^3$, where $p$ and $m$ denote positive and negative charges, see Supplementary Note 9). Even a small negative charge of $-0.25e$ already efficiently screens the larger but more distant positive charge, although the easy-plane type of anisotropy is still preserved.

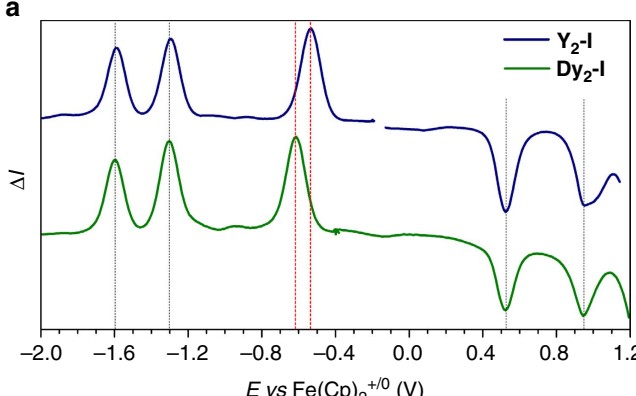

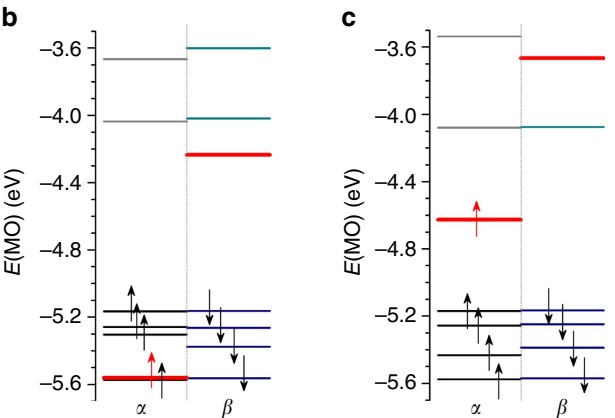

**Figure 4 | Electrochemical properties and frontier MOs.** (**a**) Square wave voltammetry of **Y₂-I** and **Dy₂-I** in 0.1 M TBABF₄/*o*-dichlorobenzene, vertical bars show positions of redox potentials (black for identical potentials in the two EMFs, red for the metal-based first reduction potentials found at $-0.52\,\mathrm{V}$ in **Y₂-I** and $-0.60\,\mathrm{V}$ **Dy₂-I**). (**b**) DFT-computed MO levels in **Y₂-I**, spin-up ($\alpha$) and spin-down ($\beta$) levels are shown separately; arrows denote electrons in occupied MOs; two components of the M–M bonding single-occupied MO are shown in red. (**c**) The same as **b** but for **La₂-I**. It is noteworthy that the energies of the M–M bonding MOs in **La₂-I** are considerably higher than those in **Y₂-I**.

When $q_m$ exceeds $-0.36e$, the sign of the CF potential acting on Dy$^{3+}$ is changed and easy-axis anisotropy develops (Fig. 5d, Supplementary Tables 8–10 and Supplementary Fig. 30). This simple model shows that the covalent bond between Dy atoms in **Dy₂-I** is not only important for the exchange coupling, but also dampens electrostatic interactions between positively charged Dy ions and enforces an easy-axis ground state for the latter.

The unique feature of the **M₂-I** system is the single-electron metal–metal bond, which results in the very strong exchange induced by the delocalized electron in the [M$^{3+}$–$e$–M$^{3+}$] system. For the hypothetical **Gd₂-I**, broken-symmetry DFT calculations predict the small $j_{\mathrm{Gd1,Gd2}}$ value of $-1.2\,\mathrm{cm}^{-1}$ and the giant $j_{\mathrm{Gd},e}$ values of 181 and 184 cm$^{-1}$ (250 and 254 K, respectively; see Supplementary Fig. 31 and Supplementary Note 10). Similar large values were predicted recently for Gd₂@C₇₉N (refs 35,36) and the EPR study of the latter revealed a $S = 15/2$ ground state, which points to the ferromagnetic coupling of all spins in the [Gd$^{3+}$–$e$–Gd$^{3+}$] system[34]. These parameters can be compared with the [Gd$^{3+}$–N$_2^{3-}$–Gd$^{3+}$] complex with a radical bridge, in which Gd ions are antiferromagnetically coupled to the electron spin of the N$_2^{3-}$ bridge with the $j_{\mathrm{Gd},e}$ value of $-27\,\mathrm{cm}^{-1}$ (ref. 16).

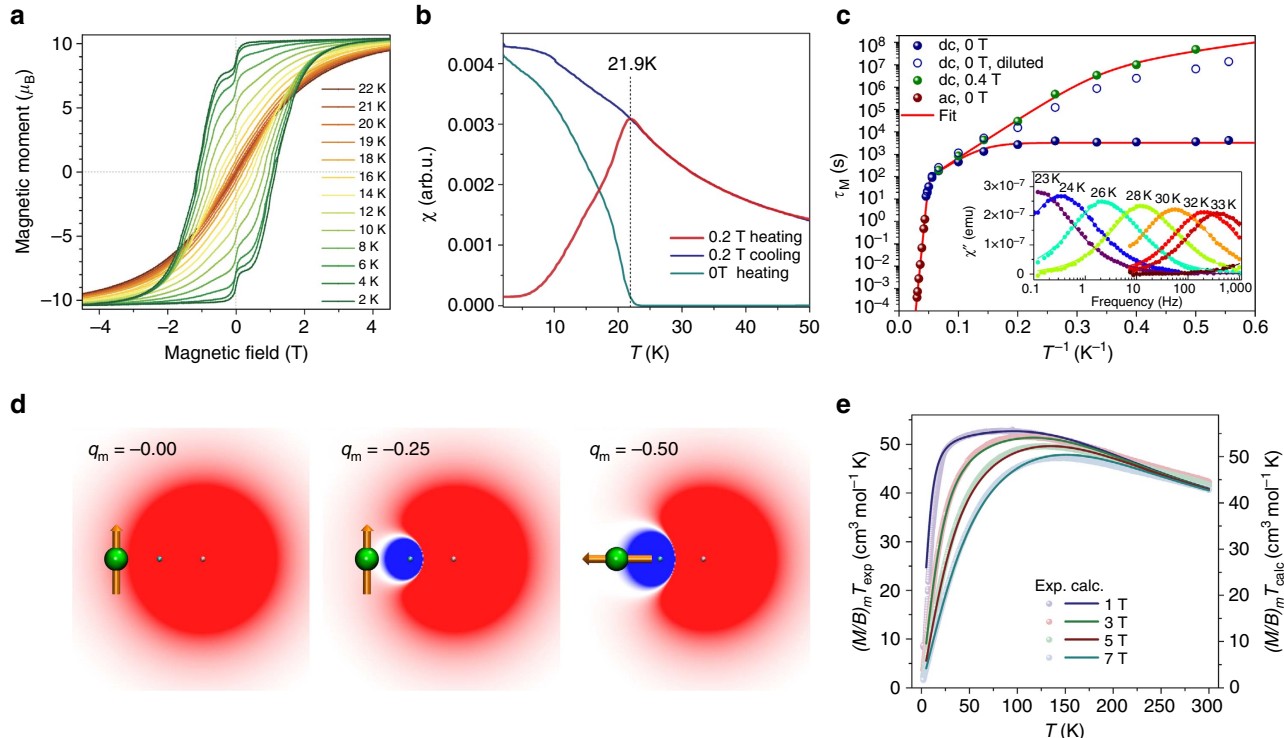

**Figure 5 | Magnetic properties of Dy₂-I. (a)** Magnetization curves measured at various temperatures with the field sweep rate of 2.9 mT s⁻¹.
**(b)** Determination of blocking temperature $T_B$: the sample is first cooled in zero-field to 1.8 K, then $\chi$ is measured in the field of 0.2 T with increasing temperature at the heating rate of 5 K min⁻¹ (red curve), then the measurement is performed at cooling down to 1.8 K (blue curve); finally, the field is turned off and decay of $\chi$ is measured at increasing temperatures (cyan curve). The vertical bar denotes $T_B$. **(c)** Magnetization relaxation times obtained from dc- and ac-measurements (dots), solid curve is a global fit using equations (1–2). The inset shows $\chi''$ values measured at different temperatures and frequencies (dots), solid lines are result of the fitting with a generalized Debye model. **(d)** Evolution of crystal field potential and the type of $Dy^{3+}$ anisotropy in the system comprising 3 + point charge placed at the distance of 3.96 Å and an additional negative charge ($q_m$) placed at the midpoint between the two Dy centres. Dy is shown as a green sphere, the arrow denotes direction of the magnetic moment. **(e)** Experimental and computed $(M/B)_m T$ curves (dots and lines, respectively) for the fields of 1, 3, 5 and 7 T. Computations were performed using the effective spin Hamiltonian in equation (4) with $j_{Dy,e} = 32$ cm⁻¹ (46 K). Experimental values are systematically smaller than theoretical one by a factor of 1.05; hence, the scales are slightly different.

The moment of 21 $\mu_B$ determined for **Dy₂-I** by SQUID magnetometry is consistent with the ferromagnetic parallel alignment of all spins in the $[Dy^{3+}–e–Dy^{3+}]$ system. More detailed information on the exchange interactions in **Dy₂-I** is revealed by the temperature and magnetic field dependence of the $(M/B)_m T$ function (Supplementary Fig. 32 and Supplementary Note 11). This function is used here instead of $\chi_m T$, because the magnetic susceptibility $\chi$ is defined as the derivative of the magnetization $M$ versus the field $B$, and the experimentally measured $M/B$ significantly deviates from $\chi$ when $B$ exceeds 1 T (Supplementary Fig. 33). In all magnetic fields studied, $(M/B)_m T$ shows a sharp increase to the maximum value of ca 55 cm³ mol⁻¹ K with increasing temperature. At higher temperatures, $(M/B)_m T$ decreases slowly reaching ca. 43 cm³ mol⁻¹ K at 300 K.

The $(M/B)_m T$ curves were simulated with a simplified version of the effective spin Hamiltonian:

$$\hat{H}_{tot} = \hat{H}_{CF(Dy1)} + \hat{H}_{CF(Dy2)} - 2j_{Dy,e}(\hat{J}_{Dy1} \cdot \hat{S}_e + \hat{J}_{Dy2} \cdot \hat{S}_e) \quad (4)$$

It is obtained from equation (3) by neglecting the $j_{Dy1,Dy2}$ constant due to its small value (Supplementary Note 10) and by considering the $j_{Dy1,e}$ and $j_{Dy2,e}$ constants to be equal to the single parameter, $j_{Dy,e}$. Similar form of the effective Hamiltonian was suggested in refs 22,49 for the description of the $[Tb–N_2^{3-}–Tb]$ complex. With the CF parameters obtained from *ab initio* calculations described above, a reasonable agreement with the experimental $(M/B)_m T$ curve measured in the field of 1 T is obtained for the $j_{Dy,e}$ constant of 30–35 cm⁻¹ (Supplementary Fig. 34). Figure 5e shows that the curves simulated for $j_{Dy,e} = 32$ cm⁻¹ (46 K) match our experimental data very well. These simulations also showed that the shape of the experimental curve requires the exchange constant to be higher than 30 cm⁻¹, and that the crystal field with the splitting of the first two CF states exceeds 200 cm⁻¹. If any of these parameters are smaller, the $\chi_m T$ and $(M/B)_m T$ functions develop a peak at low temperatures (Supplementary Figs 34 and 35) due the presence of the low-energy excited states with lower magnetic moment, whose thermal population decreases $\chi_m T$. Simulations of magnetization curves based on equation (4) and considering ferromagnetic coupling between magnetic moments of Dy centres and magnetic moment of unpaired electron perfectly reproduce experimental data measured at different temperatures (Supplementary Fig. 36 and Supplementary Note 12).

In the ground state, both Dy ions are in their $J_z = \pm 15/2$ spin states with parallel alignment (Fig. 6a). The spectrum of the spin Hamiltonian in equation (4) has two types of low-energy excited states: CF excitations of the individual $Dy^{3+}$ ions with preserved parallel alignment of the moment of the two centres and the exchange excitations, that is, the states in which the spin of one of the $Dy^{3+}$ centres is flipped (Supplementary Fig. 37, Supplementary Tables 11 and 12, and Supplementary Note 13). In the lowest-energy exchange excited states, both Dy centres still

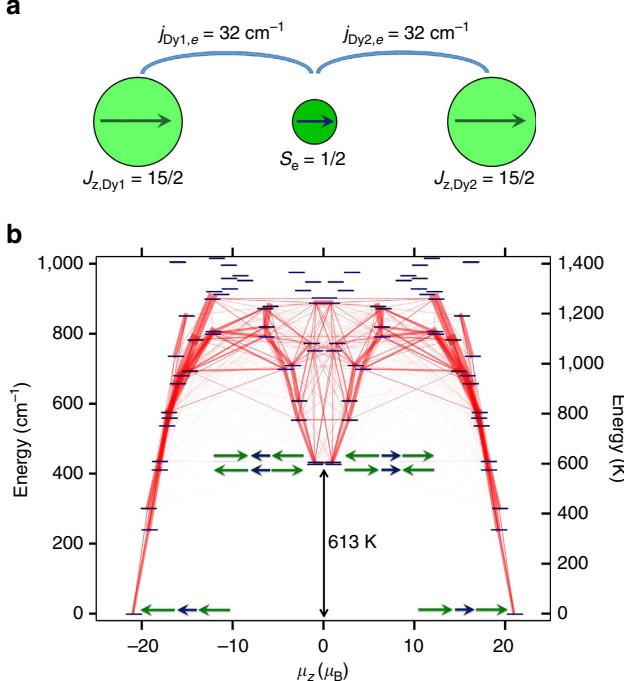

**Figure 6 | Exchange interactions and the spectrum of the spin Hamiltonian in Dy₂-I.** (**a**) Schematic description of the coupling of the magnetic moments of the $Dy^{3+}$ ions and unpaired electron spin in the ground state. (**b**) Low-energy part of the spectrum of the effective spin Hamiltonian in equation (4) with transition probabilities visualized as lines of different thickness (thicker lines correspond to higher probabilities), the $x$ axis is the projection of magnetic moment upon the main anisotropy axis, $\mu_z$. A schematic description of the spin alignment in the ground state and exchange-excited states is also shown (Dy, green arrows; single electron spin, dark blue arrow). With the $j_{Dy,e}$ constant of 32 cm$^{-1}$, the energy of the exchange states matches the Orbach barrier $U_2^{eff} = 613$ K.

have the $J_z = \pm 15/2$ spin, but now with antiparallel alignment. Variation of the $j_{Dy,e}$ constant in equation (4) from 30 to 35 cm$^{-1}$ changes the energy of these states from 390 to 450 cm$^{-1}$ (560–650 K). As a result, they are higher in energy than the lowest-energy CF-excited states found at 200–300 cm$^{-1}$ (Fig. 6b). The total spin of such states is small and it is very likely to be that relaxation of magnetization proceeds via these states, which is also confirmed by the computed transition probabilities (Fig. 6b). It is noteworthy that earlier computational studies predicted relaxation of magnetization via the first exchange-excited state in [Tb–N$_2^{3-}$–Tb] with the barrier of 299 K (ref. 49) and in Dy$_2$@C$_{79}$N with the barrier of 837 K (ref. 35). Chibotaru and colleagues[49] also developed more refined model of the exchange interactions, which showed that admixture of the CF states increases probability of the direct transition from the ground state to the first exchange-excited state. We thus conclude that the $U_2^{eff}$ value of 613 K corresponds to the lowest-energy exchange excited state in **Dy₂-I**. This value can be matched exactly by the spectrum of the spin Hamiltonian in equation (4) if the $j_{Dy,e}$ constant is set to 32 cm$^{-1}$.

The nature of the low-temperature Orbach relaxation process with the barrier of 40 K cannot be explained based on the energy spectrum of the Hamiltonian in equation (4) as the $U_1^{eff}$ value is much smaller than the energies of the excited states, whereas the $\tau_{01}$ value of 13.6 s is extremely long. We hypothesize that this Orbach-like process in fact describes relaxation via Raman mechanism assisted by low-frequency phonon modes. Studies of

the electron spin-lattice relaxation times in salts of transition metals and lanthanides showed that in the presence of a so-called localized phonon of frequency $\omega$, the rate of relaxation via the Raman mechanism can be proportional to $\exp(-\Delta\omega/k_BT)$[50–52]. Besides, a recent study of the role of phonons in under-barrier spin relaxation in SMMs revealed that an anharmonic phonon with finite linewidth may result in Orbach-like behaviour with the effective barrier corresponding to one half of the phonon frequency[53]. In endohedral fullerenes, carbon cage is rather rigid and its vibrations occur at frequencies above 200 cm$^{-1}$. Vibrational density of states at lower energies is thus quite low and is derived from vibrational modes corresponding to frustrated rotations and translations of encapsulated species. Besides, **Dy₂-I** has several low-frequency vibrations of the attached benzyl group. It is reasonable to suggest that one of these low-frequency modes and librations of the Dy$_2$ unit seem to be a particular reasonable choice, is responsible for relaxation of magnetization in **Dy₂-I** at medium-low temperatures.

## Discussion

In this work, we synthesized the single-molecule magnet with half-occupied Dy–Dy bonding orbital, Dy$_2$@C$_{80}$(CH$_2$Ph). Owing to the large ferromagnetic coupling, the endohedral [Dy$^{3+}$–$e$–Dy$^{3+}$] unit in Dy$_2$@C$_{80}$(CH$_2$Ph) behaves as a single entity with the large magnetic moment of 21 $\mu_B$. The molecule has record-high blocking temperature and a high thermal barrier of magnetization reversal of 613 K. The synthesis, although it includes a rather tedious chromatographic separation, is otherwise quite straightforward and can be performed with other lanthanides. Furthermore, a variation of the geometrical parameters and symmetry of the [Dy$^{3+}$–$e$–Dy$^{3+}$] unit may be achieved by choosing other fullerene cage sizes and isomers. In addition, Dy$_2$@C$_{80}$(CH$_2$Ph) shows a remarkable redox behaviour with stable cations and anions, and further variation of its magnetic properties can be achieved via redox chemistry. Thus, our work opens a new class of tunable air-stable single molecule magnets, whose unusual magnetic properties are due to the trapping of the unpaired electron between two lanthanides. Local 4f-based magnetic moments of individual lanthanide ions are strongly ferromagnetically coupled via unpaired electron delocalized between two metals, leading to the large net magnetic moment and high blocking temperature.

## Methods

**Synthesis.** Dy- and Y-EMFs were produced by evaporating graphite rods in the electric arc in the Krätschmer–Huffman method. The graphite rods (length 100 mm, diameter 8 mm) were packed with Dy$_2$O$_3$ or Y$_2$O$_3$ mixed with graphite (molar ratio of M:C = 1:15) and evaporated in 180 mbar helium atmosphere with the current of 100 A. The soot produced by arc vaporization was then extracted under nitrogen for 20 h by DMF at the boiling temperature of the solvent. Then, excess of benzyl bromide BrCH$_2$Ph was added to the DMF solution of extracted EMFs and heated for another 20 h at 100 °C under nitrogen protection. DMF was then evaporated with rotary evaporator and the residue was washed with methanol to remove excess of benzyl bromide. The rest was dissolved in toluene and further separated by HPLC with Buckyprep, Buckyprep-M and Buckyprep-D columns (Nacalai Tesque, Japan) as described in detail in (Supplementary Figs 4–10).

**Spectroscopic and electrochemical measurements.** Matrix-assisted laser desorption/ionization mass spectra were measured with a Bruker autoflex mass spectrometer with 1,1,4,4-tetraphenyl-1,3-butadiene as a matrix material. In a reflector mode, benzyl derivatives completely fragmented to their respective bare fullerene cores, whereas in linear mode, molecular ions could be also observed albeit with lower resolution. EPR spectra of Y-EMF solutions in DMF and toluene were measured using Bruker EMXplus spectrometer. The EPR spectra were fitted using the Easyspin programme[54]. Ultraviolet–visible–near-infrared absorption spectra were measured in toluene solution at room temperature with Shimadzu 3100 spectrophotometer. Raman spectra were recorded at 78 K on a T64000 triple spectrometer (Jobin Yvon) using 656 nm excitation wavelength of the tunable dye laser Matisse 2 (Sirah Lasertechnik) pumped by 532 nm NdYAG laser Millennia eV (Spectra-Physics). For Raman measurements, the samples were drop-casted onto

single-crystal KBr disks. Voltammetric experiments were performed in *o*-dichlorobenzene solution with $TBABF_4$ electrolyte salt in a glove box using potentiostat–galvanostat PARSTAT 4000A. A three-electrode system with a platinum working and a counter electrode and a silver wire reference electrode was used. Potentials were measured by adding ferrocene as an internal standard.

**Single-crystal X-ray diffractometry.** Crystal growth of $Dy_2@C_{80}\text{-}CH_2Ph$/$0.67(CH_3Ph)$ was accomplished by layering hexane over a solution of **Dy$_2$-I** in toluene. After the two solutions diffused together over a period of 1–2 months, small black crystals ($0.03 \times 0.03 \times 0.01$ mm$^3$) suitable for X-ray crystallographic study formed. X-ray diffraction data have been collected at 100 K on BL14.3 operated by the Joint Berlin MX Laboratory at the BESSY II electron storage ring (Berlin-Adlershof, Germany)[55] using a MAR225 CCD detector, $\lambda = 0.89429$ Å. Processing diffraction data was done with XDSAPP2.0 suite[56]. The structure was solved by direct methods and refined using all data (based on $F^2$) by SHELX 2016 (ref. 57). Hydrogen atoms were located in a difference map, added geometrically and refined with a riding model.

**Magnetometry.** Magnetization measurements were performed using a Quantum Design VSM MPMS3 magnetometer. For the powder measurements, 0.88 mg of undiluted **Dy$_2$-I** were dropcasted from $CS_2$ solution into a standard powder sample holder. For dilution measurement, the $CS_2$ solution of **Dy$_2$-I** was mixed with the solution of polystyrene in $CS_2$ to reach a 1:10,000 fullerene:polymer mass ratio. Fast evaporation of the volatile $CS_2$ gave a polymer film with dispersed fullerenes, which was then used for measurements. Long magnetization relaxation times ($> 10$ s) were determined from the measurement of magnetization decay using dc-SQUID. The sample was first magnetized to the saturation at 5 T, then the field was swept as fast as possible to zero or 0.4 T and then the decay of magnetization was followed over several hours. AC magnetometry measurements were performed using oscillation amplitude of 10 Oe (below 10 Hz) and 2 Oe (above 10 Hz) in a zero DC field.

**Calculations.** DFT optimization of the structures was performed at the PBE/TZ2P level using the Priroda[58] package. Exchange coupling parameters for **Gd$_2$-I** were obtained within the broken-symmetry approach at the PBE0/TZVP level with full-electron basis sets, scalar-relativistic DKH correction and FlipSpin routine implemented in Orca suite[59,60]. BOMDs simulation were performed at the PBE/DZVP level in CP2K code[61,62] and employed velocity Verlet algorithm with the time step of 0.5 fs and a Nosé–Hoover thermostat set at 300 K. Molecular structures, isosurfaces and BOMD trajectories were visualized using the VMD package[63]. *Ab initio* energies and wave functions of CF multiplets for the [**DyY-I**]$^-$ molecule and model systems have been calculated at the CASSCF/SO-RASSI level of theory using the quantum chemistry package MOLCAS 8.0 (ref. 45). The single ion magnetic properties and CF parameters were calculated based on *ab initio* data with the use of SINGLE_ANISO module[44]. Point-charge CF calculations were done with the McPhase code[64]. Modelling of the magnetization curves and the spin Hamiltonian solution was accomplished with the PHI programme[65].

**Data availability.** The X-ray crystallographic coordinates for the structure reported in this Article have been deposited at the Cambridge Crystallographic Data Centre, under deposition number 1519744. These data can be obtained free of charge from the Cambridge Crystallographic Data Centre via www.ccdc.cam.ac.uk/data_request.cif.

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

## Acknowledgements

We acknowledge funding by the European Research Council (ERC) under the European Union's Horizon 2020 research and innovation programme (grant agreement number 648295 'GraM3'). Computational resources were provided by the centre for Information Services and High Performance Computing (ZIH) in TU Dresden. We thank Ulrike Nitzsche for technical assistance with computational resources in IFW Dresden; Sandra Schiemenz and Frank Ziegs for the measurements of Raman and infrared spectra; Alex Beger for assistance in HPLC separation; and Sebastian Gass for the help in magnetic measurements. Manfred Weiss and Karine M. Röwer are acknowledged for the help with crystallographic measurements at BESSY.

## Author contributions

F.L. and L.S. performed synthesis, separation and crystallographic studies. Magnetometry measurements were performed by D.S.K. with help of A.K. and under guidance of T.G., A.U.B.W. and B.B. *Ab initio* calculations were performed by S.M.A. N.A.S. studied electrochemical properties. M.R. measured EPR spectra. A.A.P. initiated and led the project, performed DFT calculations, simulated magnetic properties and wrote the paper with input from all co-authors.

## Additional information

**Competing interests:** The authors declare no competing financial interests.

