## [Peer Review File · Nature Communications]

Reviewers' comments:

Reviewer #1 (Remarks to the Author):

The manuscript details the design of benzyl adducts of endohedral fullerenes containing two lanthanide centers, with a single electron metal-metal bonding interaction proposed to mediate ferromagnetic interactions between the two lanthanide centers. Such an interaction is believed to generate a large barrier to magnetic relaxation, promoting slow magnetic relaxation up to a 100-s magnetic blocking temperature of 18 K.

Concerns regarding the qualitative and incomplete nature of the magnetic data are substantial enough to mandate major revisions prior to publication. However, should the authors be able to add additional experimental evidence to the manuscript as detailed below, there is no doubt that, on the basis of novelty and relevance to a broad audience, this manuscript should be accepted to Nature Communications.

Given that the new compounds herein are reported to be isolable as both powders and single-crystals, it is unreasonable that the magnitudes of magnetization and magnetic susceptibility are reported in arbitrary units. The best confirmation of ferromagnetic coupling would be a report of molar magnetization and molar susceptibility numbers that support this magnetic ground state. With the magnetization and magnetic susceptibility data in arbitrary units, it is impossible for the reader to intuit the type of magnetic interaction occurring, and to determine whether or not magnetic impurities may be present. Computations on such electronically complex lanthanide complexes alone do not provide reliable enough evidence for a ferromagnetic ground state. A particular concern is that the current experimental data as shown in arbitrary units could potentially be replicated by a superparamagnetic ferromagnetic impurity (although this is unlikely, it must be considered). Even if non-metallated endofullerenes are isolated along with Ln₂@C₈₀ (and thus are included in the magnetic sample), metals analysis of the mixture should allow the magnetic data to be defined at least per Ln₂@C₈₀ unit.

Additionally, the cutoff of the susceptibility data at 50 K, rather than continuation of this data set up to 300 K, in addition to the lack of variable field magnetic susceptibility data (0.1 T, 1 T, etc.), makes it challenging to exclude the possibility of magnetic impurities. This data should be added. It is also standard practice in the field of molecular magnetism to report variable temperature susceptibility data in units of χ MT. Data presented in this manner should at least be included in the Supporting Information.

Finally, the authors' additional evidence for a ferromagnetic ground state seems to be Figure 3d, which, because it is not quantitative, is not sufficient evidence. The shape of M vs. H curves can easily be affected by factors such as intermolecular interactions. Using models of M vs. H with M in arbitrary units, normalized to the experimental data set to enable better fitting models, does not provide definitive evidence of a ferromagnetic ground state. As specified above, the data presentation should be altered to reflect magnetization per Ln₂@C₈₀ unit.

Additional comments:

The title is somewhat misleading about the character of the proposed metal-metal bonding electron in this system (describing the electron as "trapped between two lanthanides"). The title should be revised to reflect that a lanthanide-lanthanide bonding interaction occurs in this system (or should be revised to omit statements about the character of the electron, leaving explanation of this interaction to the abstract/manuscript).

Regarding blocking temperature, the 100-s blocking temperature definition is the only acceptable manner in which to compare single-molecule magnets from different research groups. Variable sweep rates between the zero-field-cooled versus field-cooled sweeps, and variable sweep rates

stemming from different programmed sweep rates across different magnetometers across many labs, make this method of determining blocking temperature an estimate at best. The blocking temperature should be referenced as the 100-s blocking temperature everywhere that it is noted in the abstract and manuscript, and it should be 18 K.

The EPR data of Y2-I (line 105-106) has identical parameters ($g = 1.974$ and $a(89Y) = 81.2$ G) compared to those of the endofullerene complex Y2@C79N (ref 31). A comment should be included on this similarity, and whether this similarity is reasonable, including a re-reference of ref. 31. How can the authors be sure that they didn't simply generate Y2@C79N instead of Y2@C80?

It is not clear what relaxation components (Orbach, Raman, etc.) were used to generate the fit of the relaxation times for diluted Dy2@C80 (Figure 3c). A statement should be added to clarify this in the manuscript or figure caption. It seems like the second, smaller barrier Orbach process is not necessary for the diluted Dy2@C80 fit.

Also regarding the Ueff,2 process, it seems like it may be possible to fit this part of the Arrhenius plot using a Raman process with a low value of n , instead of with a low energy barrier Orbach process. Did the authors try this? Perhaps a comment on this could be added to the SI? The τ_0 of 13.6 s does indeed seem unreasonable, as the authors note, such that fitting with a Raman process with low n would seem preferable, if possible.

The sentence "experimental data on lanthanide-copper system show $J(Gd,e)$ values are usually 3-4 times higher than $J(Dy,e)$ " (line 211-212) should be removed. The mechanism of magnetic coupling in Gd-Cu complexes is likely very different from that occurring for the Dy2@C80 complex, and thus this does not seem like a reasonable comparison.

If possible, some type of yield of the Ln2@C80 complexes should be reported in the Supporting Information (How much Dy2@C80 is present in the final fraction A-4-2-3 from what initial amount of Dy2O3 and graphite rods? What mass of Dy2@C80 single crystals did the authors obtain, from how much starting crystallization material?)

Reviewer #2 (Remarks to the Author):

The authors present magnetic relaxation properties for some Dy2 endohedral metal fullerenes with a single electron bond.

Overall I find the paper lacking in evidence and the modelling very poorly explained and justified. Furthermore, the script is almost unintelligible, even for someone familiar with the field. This paper needs major revisions, but afterwards could be suitable for Nat. Commun.

Specific comments:

Figure S2 top is poorly interpreted. What is the large feature at ca. 3490 G? Are the g -values for the different isomers the same? If so, why are the hyperfine splittings of the different isomers all shown with reference to the low-field $g \sim 2.0005$ feature? Hyperfine splitting should be measured from the central peak for each isomer. Furthermore, I cannot see the 1:2:1 intensity as claimed. Figure S2 bottom has no interpretation whatsoever, so they could claim anything. I see no evidence to support their claim. Both these spectra need to be fully modelled with all components and parameters reported.

The broad optical spectra shown in Figure S13 does not prove the two compounds are isostructural, and while the Raman data looks good, is there any powder XRD data to confirm this

claim?

Figure 3a: I can definitely see an open hysteresis to 18 K, but to say it's open until 22 K is a stretch. Perhaps a figure in the SI showing just the higher temperatures to make it clear?

Line 136: On what basis has the $\tau = 100$ s temperature been calculated?

When fitting the DC relaxation data, what stretch parameters were obtained? In fact, the results of these measurements and fits should be reported in the SI as tables and figures.

The relaxation time data has been fit with equation 1; what is the origin/basis/explanation for the two Orbach terms? This is mentioned in the SI, but needs to be addressed in the script. While the second Orbach term has parameters consistent with literature on Dy SMMs, those for the first Orbach term are completely unphysical. The pre-factor is related to the spin-phonon collision time, which will not be ca. 14 s! I think it is far more likely that the relaxation below 20 K in both cases is Raman-like and below 5 K for the non-dilute zero-field case it's QTM. The modelling of the kink at the lowest temperatures in the dilute and 0.4 T pure data is likely not a real feature of the data. I would like to see this remodelled without the "low-barrier" Orbach term and using the Raman term to provide this curving profile. If the authors can show that this is not feasible, then perhaps I will be more inclined to believe their explanation in the SI.

What model was employed for the calculation of the Dy-e-Dy magnetization curves? How was the exchange with the radical treated? What was the local anisotropy (and relative symmetry) imposed on the Dy ions? This is explained in very confusing detail in the SI, but need to be succinctly explained in the script. Furthermore, with no absolute values for the magnetic measurements, how is it possible that accurate comparisons can be made to determine the absolute magnetic moments? This is simply nonsense!

Line 207: at no point is the χT or μ_{eff} shown as a function of temperature and there are no absolute values on any of the magnetization plots; thus, there is actually no data to support a ferromagnetic interaction. I believe there is slow relaxation behaviour, but I see no evidence on the nature or strength of the exchange interaction. Certainly, Figure 3d and the simulations do not provide this evidence.

Table S3: Given that the g-tensors for the higher energy Kramers doublets are not axial, it is completely incorrect to associate these energy levels with pure mJ states.

CASSCF calculations: Given the (supposed) strong exchange, it should actually be fairly easy to get a full CASSCF model for the Dy-e-Dy EMF, using a General Active Space. In this way, the magnetic interaction would fall out directly from the calculations and they would also provide evidence on the nature of the coupled ground state.

Discussion of (large) U_{eff} barrier: The discussion of this in the SI is very convoluted and unclear. This section needs to be reduced to the main points and discussed in the script clearly.

Reviewer #3 (Remarks to the Author):

The manuscript by Liu et al. is reporting original and very interesting results, combining in clear and accessible manuscript a very detailed chemical and physical characterization of the novel endohedral fullerene complexes.

The manuscript fulfills all requirements for publication in Nature Communications and should be promptly accepted because it represents a significant advancement in the field of molecular nanomagnetism. The potentiality of the approach developed by the corresponding authors and his group are really astonishing.

I have only very minor points that the authors could take into account to improve their manuscript.

- a) The authors have measured the solution EPR spectrum of the Y₂ derivative only at room temperature. Why? They could have gone at low temperature to detect the anisotropy of the hyperfine coupling (they say that the inner atoms do not rotate at low temperature). The observation of a very anisotropic HF coupling would have been the smoking gun that the e⁻ is not delocalized on the cage but actually between the two Y³⁺ ions.
- b) In the supplementary information they describe the analysis they have performed on the temperature dependence of the relaxation time. They could transfer part of this text in the main article and provide also the parameters of the fit of the data taken in a dc field, to show if they are consistent with their theory. The two barriers and Raman exponent should not change with the field, only the tunneling rate.

Reviewer #4 (Remarks to the Author):

Liu et al. report the successful capture and structure elucidation of two elusive endohedral fullerenes M₂@C₈₀ (M = Dy, Y) via a facile synthesis of their air-stable derivatives M₂@C₈₀(CH₂Ph). So far dimetallofullerenes M₂@C₈₀ were synthesized only for La, Ce, and Pr, while M₂@C₈₀ molecules with Y and medium-to small size lanthanides (Gd-Lu) have never been isolated in any significant amounts. This work proves that Y (Dy)-based M₂@C₈₀ molecules are actually formed in the synthesis in reasonable amounts, and their "absence" in the products had been caused by their low chemical stability. This work opens the way to the full-scale exploration of these elusive di-metallofullerenes. Moreover, the authors show that such dimetallofullerenes have unusual magnetic properties caused by their peculiar electronic structure, and Dy₂@C₈₀(CH₂Ph) is shown to be a single molecule magnet with record high blocking temperature. Based on the high novelty of this work, I recommend to publish it in Nature Commun., but minor revisions are required.

1. The authors mention that Dy ions in the crystal structure are disordered, but did not discuss the details. Besides, the packing of the molecules in the crystal is not discussed at all. I am wondering whether the packing pattern of the molecules in the crystal would influence their magnetic properties. The authors should give more details on these questions.
2. The electrochemical properties of these two new structures and relation to the "metal-metal bond" are quite interesting. I believe it would attract an interest in the fullerene community and beyond. However, the discussion of electrochemical analysis in the main text is too limited, and I recommend to move related discussion from S.I. to the main text as the space permits. Besides, comparison to the analogous La₂@C₈₀ derivative reported recently by Lu et al. should be also made and discussed.
3. Discussion of the spectroscopic characterization (UV-vis and Raman spectra) of the two structures in the main text is too rudimentary. Please provide more detailed analysis why these particular techniques are used here and what they tell us about the new molecules.

We would like to thank the reviewers for their critical comments, which encouraged us to make new measurements and perform more thorough and reliable analysis of the measured data. The presentation of the magnetic properties is largely rewritten.

The major changes in the revised version include:

- 1) The absolute values of magnetic moment and magnetic susceptibility have been determined
- 2) $\chi_m T$ (more precise, $(M/B)_m T$) curves have been measured in different fields from 0.2 T and up to 7 T and simulated using effective spin Hamiltonian, which allowed determination of the exchange coupling parameters.
- 3) With the *ab initio* computed CF parameters and with the newly determined coupling constant, better understanding of the spectrum of Hamiltonian has been achieved and reliable interpretation of the high-barrier Orbach process could be obtained.
- 4) A number of further improvements and corrections is implemented, including EPR spectra of frozen solution and more detailed analysis of electrochemical data.

Detailed point-to-point responses to Reviewers' comments are given below.

Reviewer #1 (Remarks to the Author):

The manuscript details the design of benzyl adducts of endohedral fullerenes containing two lanthanide centers, with a single electron metal-metal bonding interaction proposed to mediate ferromagnetic interactions between the two lanthanide centers. Such an interaction is believed to generate a large barrier to magnetic relaxation, promoting slow magnetic relaxation up to a 100-s magnetic blocking temperature of 18 K.

Concerns regarding the qualitative and incomplete nature of the magnetic data are substantial enough to mandate major revisions prior to publication. However, should the authors be able to add additional experimental evidence to the manuscript as detailed below, there is no doubt that, on the basis of novelty and relevance to a broad audience, this manuscript should be accepted to Nature Communications.

Given that the new compounds herein are reported to be isolable as both powders and single-crystals, it is unreasonable that the magnitudes of magnetization and magnetic susceptibility are reported in arbitrary units. The best confirmation of ferromagnetic coupling would be a report of molar magnetization and molar susceptibility numbers that support this magnetic ground state. With the magnetization and magnetic susceptibility data in arbitrary units, it is impossible for the reader to intuit the type of magnetic interaction occurring, and to determine whether or not magnetic impurities may be present. Computations on such electronically complex lanthanide complexes alone do not provide reliable enough evidence for a ferromagnetic ground state. A particular concern is that the current experimental data as shown in arbitrary units could potentially be replicated by a superparamagnetic ferromagnetic impurity (although this is unlikely, it must be considered). Even if non-metallated endofullerenes are isolated along with Ln₂@C₈₀ (and thus are included in the magnetic sample), metals analysis of the mixture should allow the magnetic data to be defined at least per Ln₂@C₈₀ unit.

Magnetization measurements shown in Figure 3a (revised manuscript) have been performed for the sample of the known (albeit small) mass, 0.88 mg. The revised version hence provides the magnetic moment per molecule (which amounts to $20.9 \pm 0.3 \mu_B$ per molecule; see Supporting Information S11.1). Susceptibility data (see below) are also provided in molar units. The sample does not contain non-metallated (aka empty) fullerenes or any other fullerenes.

Additionally, the cutoff of the susceptibility data at 50 K, rather than continuation of this data set up to 300 K, in addition to the lack of variable field magnetic susceptibility data (0.1 T, 1 T, etc.), makes it challenging to exclude the possibility of magnetic impurities. This data should be added. It is also standard practice in the field of molecular magnetism to report variable temperature susceptibility data in units of χ MT. Data presented in this manner should at least be included in the Supporting Information.

In the revised version, we provide $\chi_m T$ (and more precise, $(M/B)_m T$) data measured up to room temperature and in different fields, from 0.2 T to 7 T (Figure 4e, Supporting Information S11.12).

Finally, the authors' additional evidence for a ferromagnetic ground state seems to be Figure 3d, which, because it is not quantitative, is not sufficient evidence. The shape of M vs. H curves can easily be affected by factors such as intermolecular interactions. Using models of M vs. H with M in arbitrary units, normalized to the experimental data set to enable better fitting models, does not provide definitive evidence of a ferromagnetic ground state. As specified above, the data presentation should be altered to reflect magnetization per Ln₂@C80 unit.

In the revised version, we used $\chi_m T$ and $(M/B)_m T$ measurements to compare to the simulated data. These simulations not only confirmed the FM coupling but also enabled us to estimate the coupling constants (which was not possible in the original version). Besides, M(H) curves were measured at different temperatures up to 100 K and compared to the computed ones.

Additional comments:

The title is somewhat misleading about the character of the proposed metal-metal bonding electron in this system (describing the electron as “trapped between two lanthanides”). The title should be revised to reflect that a lanthanide-lanthanide bonding interaction occurs in this system (or should be revised to omit statements about the character of the electron, leaving explanation of this interaction to the abstract/manuscript).

We believe that the current title expresses the peculiarity of the system and does it in a language clear not only to a chemical audience (familiar with the terms like “single-electron bond”) but also for a more general audience. The details of the bonding situation are then clarified in the abstract.

Regarding blocking temperature, the 100-s blocking temperature definition is the only acceptable manner in which to compare single-molecule magnets from different research groups. Variable sweep rates between the zero-field-cooled versus field-cooled sweeps, and variable sweep rates stemming from different programmed sweep rates across different magnetometers across many labs, make this method of determining blocking temperature an estimate at best. The blocking temperature should be referenced as the 100-s blocking temperature everywhere that it is noted in the abstract and manuscript, and it should be 18 K.

The $T_{B(100)}$ has been referred to in the abstract. Since T_B measured as the peak in the susceptibility is very often used in the literature, we decided to keep the value(s) in the manuscript. But, encouraged by the comment of the Reviewer, we measured T_B values at different temperature sweep rates and indeed found rather large variation of this parameter (Supporting Information S11.3). Hence, in the revised version of

the manuscript, we list the values measured for several sweep rates. We hope that this situation will attract attention of other researchers to this problem as temperature sweep rates are often not reported when T_B is measured.

The EPR data of Y2-I (line 105-106) has identical parameters ($g = 1.974$ and $a(89Y) = 81.2$ G) compared to those of the endofullerene complex Y2@C79N (ref 31). A comment should be included on this similarity, and whether this similarity is reasonable, including a re-reference of ref. 31. How can the authors be sure that they didn't simply generate Y2@C79N instead of Y2@C80?

The sample preparation and characterization (mass-spectrometry, for instance – Supporting Information S10, Figures S10) ensures that Y2-I is not Y2@C79N, but Y2@C80(CH2Ph). We fitted experimental spectrum using EasySpin code and obtained slightly different g -factor and coupling constant (originally, the values were obtained from the central peak position and using the distance between the peaks). We added a note that close similarity of the parameters is reasonable as both molecules have the same spin system (single-electron bond between two Y atoms)

It is not clear what relaxation components (Orbach, Raman, etc.) were used to generate the fit of the relaxation times for diluted Dy2@C80 (Figure 3c). A statement should be added to clarify this in the manuscript or figure caption. It seems like the second, smaller barrier Orbach process is not necessary for the diluted Dy2@C80 fit.

We did not include the data for the diluted sample (empty circles in Figure 3c) in the fit. The fit was performed for non-diluted sample (closed blue, green, and brown circles), and zero-field and in-filed measurements were fitted with the same set of parameters.

Also regarding the Ueff,2 process, it seems like it may be possible to fit this part of the Arrhenius plot using a Raman process with a low value of n , instead of with a low energy barrier Orbach process. Did the authors try this? Perhaps a comment on this could be added to the SI? The τ_0 of 13.6 s does indeed seem unreasonable, as the authors note, such that fitting with a Raman process with low n would seem preferable, if possible.

We tried to fit the data without the low-barrier Orbach process, and could not obtain satisfactory results. The comment is added to the manuscript, and the fit is included in SI (Supporting Information S11.7, Figure S28)

The sentence “experimental data on lanthanide-copper system show $J(\text{Gd,e})$ values are usually 3-4 times higher than $J(\text{Dy,e})$ ” (line 211-212) should be removed. The mechanism of magnetic coupling in Gd-Cu

complexes is likely very different from that occurring for the Dy₂@C₈₀ complex, and thus this does not seem like a reasonable comparison.

We followed suggestion of the reviewer. In fact, the use of $(M/B)_m T$ measurements allowed estimation of the Dy-electron coupling constant directly for our data.

If possible, some type of yield of the Ln₂@C₈₀ complexes should be reported in the Supporting Information (How much Dy₂@C₈₀ is present in the final fraction A-4-2-3 from what initial amount of Dy₂O₃ and graphite rods? What mass of Dy₂@C₈₀ single crystals did the authors obtain, from how much starting crystallization material?)

The data on the yield is added to the SI (Supporting information S6). The mass of the crystal is tiny because of its small size ($0.03 \times 0.03 \times 0.01 \text{ mm}^3$). The procedure gave several crystals.

Reviewer #2 (Remarks to the Author):

The authors present magnetic relaxation properties for some Dy₂ endohedral metal fullerenes with a single electron bond.

Overall I find the paper lacking in evidence and the modelling very poorly explained and justified. Furthermore, the script is almost unintelligible, even for someone familiar with the field. This paper needs major revisions, but afterwards could be suitable for Nat. Commun.

Specific comments:

Figure S2 top is poorly interpreted. What is the large feature at ca. 3490 G?

The large peak with $g=2.0005$ is presumably originating from anions of empty fullerenes and/or oxidized form of DMF (we added this comment in the revised version of SI). The peak is not present in the sample after HPLC separation, therefore its nature was not of the major concern.

Are the g -values for the different isomers the same?

The g -factors for three main species are different, as shown in the revised Figure S2.

If so, why are the hyperfine splittings of the different isomers all shown with reference to the low-field $g \sim 2.0005$ feature? Hyperfine splitting should be measured from the central peak for each isomer.

In fact, hyperfine splittings are shown with respect to the central line of each triplet, but the signals have accidentally coinciding low-field peaks.

Furthermore, I cannot see the 1:2:1 intensity as claimed.

We show an absorption spectrum in addition to the first-derivative in Figures S2 and S3. While we cannot measure the ratio of each individual component because of their strong overlap, the net ratio of the low-field to middle-field to high-field components of the triplet signals is very close to 1:2:1.

Figure S2 bottom has no interpretation whatsoever, so they could claim anything. I see no evidence to support their claim. Both these spectra need to be fully modelled with all components and parameters reported.

Showing the spectrum of the mixture after reaction with Br-benzyl (Figure S3 in the revised version) we claim that the species with large hfc couplings (characteristic for the single-electron Y-Y bond) are preserved in the sample after reaction. This statement can hardly be questioned. We added the following comment in the supporting information:

“Note that each of the three major $Y_2@C_{2n}^-$ anions present in the DMF extract before the reaction may give several isomers of the benzyl adducts. Besides, Y atoms in such adducts may be non-equivalent. Combination of these two factors leads to a complex EPR spectrum of the mixture of $Y_2@C_{2n}(CH_2Ph)$ derivatives (Figure S3). Despite the complexity, the spectrum clearly shows that $Y_2@C_{2n}(CH_2Ph)$ derivatives do have large ^{89}Y hfc constants of 65-80 G, which proves that the single-occupied Y-Y bonding MO is preserved in $Y_2@C_{2n}(CH_2Ph)$ derivatives. Note also that the sharp peak near 3490 G indicates that non-Y organic radicals are also formed in the reaction. However, it is not practical and not essential for this work

to give a detailed interpretation of the whole complex spectrum; subsequent HPLC separation of this mixture gave the pure **Y₂-I** compound, which has simple triplet spectrum and does not contain any other radical impurities (see Figure 2 in the manuscript).”

The broad optical spectra shown in Figure S13 does not prove the two compounds are isostructural, and while the Raman data looks good, is there any powder XRD data to confirm this claim?

Endohedral fullerenes usually give strongly disordered crystals, which may also include some poorly controllable amount of solvent molecules. Therefore, spectroscopic data (UV-Vis, Raman, and IR spectroscopy) provide much better evidence for the similarity of the molecular structures than the XRD. It has been reported many times (for instance, see our review in *Chem. Rev.* **2013**, *113*, 5989) that fullerenes with different metals but identical carbon cage structures in the same formal charges states exhibit very similar absorption and vibrational spectra. In the revised version, we added IR spectra (to already measured UV-Vis and Raman) and again found very close similarity between the spectra of **Y₂-I** and **Dy₂-I**. These spectra provide compelling evidence that **Y₂-I** and **Dy₂-I** have identical molecular structure (in the terms of the carbon cage isomer and position of the external group).

Figure 3a: I can definitely see an open hysteresis to 18 K, but to say it's open until 22 K is a stretch. Perhaps a figure in the SI showing just the higher temperatures to make it clear?

We added Figure S21 in SI to show that opening can be seen up to 21 K (at 22 K the hysteresis closes – at least at the sweep rate of 2.9 mT/s used in our work).

Line 136: On what basis has the $\tau = 100$ s temperature been calculated?

The temperature was calculated from the Figure 3c. Using the data fitting curve for zero-field relaxation data, we obtained the temperature of 17.96 K for 100-s relaxation time.

When fitting the DC relaxation data, what stretch parameters were obtained? In fact, the results of these measurements and fits should be reported in the SI as tables and figures

We added tables listing all determined time and stretch parameters (Table S2, S3) as well as the values determined in ac measurements (Table S4, S5)

The relaxation time data has been fit with equation 1; what is the origin/basis/explanation for the two Orbach terms? This is mentioned in the SI, but needs to be addressed in the script. While the second Orbach term has parameters consistent with literature on Dy SMMs, those for the first Orbach term are completely unphysical. The pre-factor is related to the spin-phonon collision time, which will not be ca. 14 s! I think it is far more likely that the relaxation below 20 K in both cases is Raman-like and below 5 K

for the non-dilute zero-field case it's QTM. The modelling of the kink at the lowest temperatures in the dilute and 0.4 T pure data is likely not a real feature of the data. I would like to see this remodelled without the "low-barrier" Orbach term and using the Raman term to provide this curving profile. If the authors can show that this is not feasible, then perhaps I will be more inclined to believe their explanation in the SI.

We tried to fit the data w/o the low-barrier Orbach process (Supporting Information S11.7, Figure S28) but could not obtain satisfactory results. Possible explanations for the low-barrier Orbach process are now discussed in the manuscript in the last paragraph before conclusions. We think that this process may correspond to either Raman relaxation with local mode (which will give Orbach-like dependence with the "barrier" equal to the phonon frequency) or to the anharmonic phonon relaxation as recently discussed in detail in *Nat Commun* 2017, **8**: 14620 (this process may also result in Arrhenius behaviour with the barrier equal to the half of the phonon frequency).

What model was employed for the calculation of the Dy-e-Dy magnetization curves? How was the exchange with the radical treated? What was the local anisotropy (and relative symmetry) imposed on the Dy ions? This is explained in very confusing detail in the SI, but need to be succinctly explained in the script.

We have completely rewritten the section about the modelling of the data. The effective spin Hamiltonian (with its crystal field and exchange terms) is shown explicitly and each component is discussed in the manuscript as well as in the revised SI. In brief, CF parameters are computed *ab initio*, exchange parameter is now determined from $(M/B)_mT$ data (Supporting Information S11.12).

Furthermore, with no absolute values for the magnetic measurements, how is it possible that accurate comparisons can be made to determine the absolute magnetic moments? This is simply nonsense!

Absolute values are now given for magnetization curves in Figure 3a, determination of magnetic moment is discussed in Supporting Information S11.1 and S11.13

Line 207: at no point is the χT or μ_{eff} shown as a function of temperature and there are no absolute values on any of the magnetization plots; thus, there is actually no data to support a ferromagnetic interaction. I believe there is slow relaxation behaviour, but I see no evidence on the nature or strength of the exchange interaction. Certainly, Figure 3d and the simulations do not provide this evidence.

Figure 4e now includes $(M/B)_mT$ data measured in different fields and in absolute values. This function was found to be very sensitive to the exchange parameters (Supporting Information S11.12, Figure S34).

Table S3: Given that the g-tensors for the higher energy Kramers doublets are not axial, it is completely incorrect to associate these energy levels with pure mJ states.

The table were corrected

CASSCF calculations: Given the (supposed) strong exchange, it should actually be fairly easy to get a full CASSCF model for the Dy-e-Dy EMF, using a General Active Space. In this way, the magnetic interaction would fall out directly from the calculations and they would also provide evidence on the nature of the coupled ground state.

Unfortunately, we did not find such computations feasible with available computational resources.

Discussion of (large) U_{eff} barrier: The discussion of this in the SI is very convoluted and unclear. This section needs to be reduced to the main points and discussed in the script clearly.

In the revised version, with better determined coupling parameters, we have been able to construct and discuss the spectrum of the Hamiltonian and showed that the barrier corresponds to the exchange-excited state, in which the spin on one of the Dy centers is flipped.

Reviewer #3 (Remarks to the Author):

The manuscript by Liu et al. is reporting original and very interesting results, combining in clear and accessible manuscript a very detailed chemical and physical characterization of the novel endohedral fullerene complexes.

The manuscript fulfills all requirements for publication in Nature Communications and should be promptly accepted because it represents a significant advancement in the field of molecular nanomagnetism. The potentiality of the approach developed by the corresponding authors and his group are really astonishing.

I have only very minor points that the authors could take into account to improve their manuscript.

a) The authors have measured the solution EPR spectrum of the Y₂ derivative only at room temperature. Why? They could have gone at low temperature to detect the anisotropy of the hyperfine coupling (they say that the inner atoms do not rotate at low temperature). The observation of a very anisotropic HF coupling would have been the smoking gun that the e⁻ is not delocalized on the cage but actually between the two Y³⁺ ions.

The measurements for the frozen solution have been performed and anisotropic g- and a-tensors were obtained (Figure 2 in the revised manuscript)

b) In the supplementary information they describe the analysis they have performed on the temperature dependence of the relaxation time. They could transfer part of this text in the main article and provide also the parameters of the fit of the data taken in a dc field, to show if they are consistent with their theory. The two barriers and Raman exponent should not change with the field, only the tunneling rate.

We changed description of the analysis in the text to clarify these points. In fact, we used in-field and zero-field data together in obtaining one set of parameters describing both sets of data.

Reviewer #4 (Remarks to the Author):

Liu et al. report the successful capture and structure elucidation of two elusive endohedral fullerenes $M_2@C_{80}$ ($M = Dy, Y$) via a facile synthesis of their air-stable derivatives $M_2@C_{80}(CH_2Ph)$. So far dimetallofullerenes $M_2@C_{80}$ were synthesized only for La, Ce, and Pr, while $M_2@C_{80}$ molecules with Y and medium-to small size lanthanides (Gd-Lu) have never been isolated in any significant amounts. This work proves that Y (Dy)-based $M_2@C_{80}$ molecules are actually formed in the synthesis in reasonable amounts, and their “absence” in the products had been caused by their low chemical stability. This work opens the way to the full-scale exploration of these elusive di-metallofullerenes. Moreover, the authors show that such dimetallofullerenes have unusual magnetic properties caused by their peculiar electronic structure, and $Dy_2@C_{80}(CH_2Ph)$ is shown to be a single molecule magnet with record high blocking temperature. Based on the high novelty of this work, I recommend to publish it in Nature Commun., but minor revisions are required.

1. The authors mention that Dy ions in the crystal structure are disordered, but did not discuss the details. Besides, the packing of the molecules in the crystal is not discussed at all. I am wondering whether the packing pattern of the molecules in the crystal would influence their magnetic properties. The authors should give more details on these questions.

We discussed the disorder and packing of the molecules in the crystal in more details (Supporting Information S7). Dilution measurements showed that intermolecular interaction affect QTM-like regime, but other temperature regimes remain unaffected. Therefore, intermolecular interactions (and packing) do not affect relaxation rates except for the QTM regime.

2. The electrochemical properties of these two new structures and relation to the “metal-metal bond” are quite interesting. I believe it would attract an interest in the fullerene community and beyond. However, the discussion of electrochemical analysis in the main text is too limited, and I recommend to move related discussion from S.I. to the main text as the space permits. Besides, comparison to the analogous $La_2@C_{80}$ derivative reported recently by Lu et al. should be also made and discussed.

Discussion of electrochemical data in the manuscript is expanded according to these recommendations, comparison to the $La_2@C_{80}$ derivative is given.

3. Discussion of the spectroscopic characterization (UV-vis and Raman spectra) of the two structures in the man text is too rudimentary. Please provide more detailed analysis why these particular techniques are used here and what they tell us about the new molecules.

Discussion of the spectroscopic date has been added to Supporting Information S9.

REVIEWERS' COMMENTS:

Reviewer #1 (Remarks to the Author):

The authors have provided an excellent revised manuscript that has satisfied all of this reviewer's prior concerns. Some minor changes are listed below that should be made prior to manuscript publication.

Page 11, line 273: The exchange Hamiltonian of the type in equation 3, which assumes that in the case of strong coupling between an isotropic spin and anisotropic lanthanide ions, the total angular momentum J of each lanthanide ion should be used to model the exchange interaction, was recently used to describe a somewhat similar complex ($\text{Ln}_2(\mu\text{-N}_2\text{O}_3)$); two anisotropic lanthanides bridged by an isotropic $S = 1/2$ in the reference:

Vieru, V.; Iwahara, N.; Ungur, L.; Chibotaru, L. F. Giant Exchange Interaction in Mixed Lanthanides. *Sci. Rep.* 2016, 6, 24046.

as well as in ref. 22. These references should be cited (or re-cited in the case of ref 22) in this section of the main manuscript, immediately preceding equation 3.

Typos:

Page 7, line 151 (Figure 3 caption): "meal-based" should be "metal-based"

Page 11, Line 262: "-27.7 cm^{-1} " should be "-27 cm^{-1} "

Page 12, Line 302: "anharmonic phonons" should be "anharmonic phonon"

Page 12, Line 303: "the Orbach" should just be "Orbach"

Reviewer #2 (Remarks to the Author):

The authors have revised their manuscript and addressed most of the concerns of the referees. This manuscript is now suitable for publication.

- In the first paragraph about magnetic interactions, the hysteresis figure is S21 not S20 "shown in Figure 4a and S20"

- The authors state "Taking into account the random orientation of crystallites in the powder, the real magnetic moment amounts to $20.9 \pm 0.3 \mu\text{B}$ per molecule." but of course this is along a specific direction as the molecules are magnetically anisotropic. There is nothing 'un-real' about $10.5 \mu\text{B}$ per molecule; it is the powder integration of the anisotropic magnetic moment.

- A comment/thought on the origin of the much stronger and ferromagnetic Gd-e^- exchange of ca. $+180 \text{ cm}^{-1}$ vs. -30 cm^{-1} for $\text{Gd-N}_2\text{O}_3$ would be useful.

Reviewer #4 (Remarks to the Author):

The authors have addressed all of my comments clearly, thus the manuscript can be accepted in its present form.

Reviewer #1 (Remarks to the Author):

The authors have provided an excellent revised manuscript that has satisfied all of this reviewer's prior concerns. Some minor changes are listed below that should be made prior to manuscript publication.

Page 11, line 273: The exchange Hamiltonian of the type in equation 3, which assumes that in the case of strong coupling between an isotropic spin and anisotropic lanthanide ions, the total angular momentum J of each lanthanide ion should be used to model the exchange interaction, was recently used to describe a somewhat similar complex ($\text{Ln}_2(\mu\text{-N}_2\text{O}_3)$; two anisotropic lanthanides bridged by an isotropic $S = 1/2$) in the reference:

Vieru, V.; Iwahara, N.; Ungur, L.; Chibotaru, L. F. Giant Exchange Interaction in Mixed Lanthanides. *Sci. Rep.* 2016, 6, 24046.

as well as in ref. 22. These references should be cited (or re-cited in the case of ref 22) in this section of the main manuscript, immediately preceding equation 3.

Following Reviewer's suggestion, Ref. 22 as well as the paper by Vieru et al. were cited when discussing the spin Hamiltonian in Equation (4) (former Equation 3):

"Similar form of the effective Hamiltonian was suggested in Refs. 22 and 49 for the description of the $[\text{Tb}-\text{N}_2\text{O}_3-\text{Tb}]$ complex."

Typos:

Page 7, line 151 (Figure 3 caption): "meal-based" should be "metal-based"

Page 11, Line 262: "-27.7 cm^{-1} " should be "-27 cm^{-1} "

Page 12, Line 302: "anharmonic phonons" should be "anharmonic phonon"

Page 12, Line 303: "the Orbach" should just be "Orbach"

These and some other typos were corrected

Reviewer #2 (Remarks to the Author):

The authors have revised their manuscript and addressed most of the concerns of the referees. This manuscript is now suitable for publication.

- In the first paragraph about magnetic interactions, the hysteresis figure is S21 not S20 "shown in Figure 4a and S20"

The number of the figure was adjusted

- The authors state "Taking into account the random orientation of crystallites in the powder, the real magnetic moment amounts to $20.9 \pm 0.3 \mu\text{B}$ per molecule." but of course this is along a specific

direction as the molecules are magnetically anisotropic. There is nothing 'un-real' about 10.5 μ_B per molecule; it is the powder integration of the anisotropic magnetic moment.

The phrase in question was changed to "the magnetic moment along the main magnetization axis amounts to $20.9 \pm 0.3 \mu_B$ per molecule"

- A comment/thought on the origin of the much stronger and ferromagnetic Gd-e⁻ exchange of ca. +180 cm^{-1} vs. -30 cm^{-1} for Gd-N₂³⁻ would be useful.

Such a comment would be too premature and speculative at this moment. The model to explain this phenomenon is under development now.